# The Etiology, Antibiotic Therapy and Outcomes of Bacteremic Skin and Soft-Tissue Infections in Onco-Hematological Patients

**DOI:** 10.3390/antibiotics12121722

**Published:** 2023-12-13

**Authors:** Valeria Castelli, Enric Sastre-Escolà, Pedro Puerta-Alcalde, Leyre Huete-Álava, Júlia Laporte-Amargós, Alba Bergas, Mariana Chumbita, Mar Marín, Eva Domingo-Domenech, Ana María Badia-Tejero, Paula Pons-Oltra, Carolina García-Vidal, Jordi Carratalà, Carlota Gudiol

**Affiliations:** 1Infectious Diseases Department, Bellvitge University Hospital, Bellvitge Biomedical Research Institute (IDIBELL), University of Barcelona, L’Hospitalet de Llobregat, 08907 Barcelona, Spain; valeria.castelli@unimi.it (V.C.); j.laporte@bellvitgehospital.cat (J.L.-A.); abergas@bellvitgehospital.cat (A.B.); abadia@bellvitgehospital.cat (A.M.B.-T.); ppons@bellvitgehospital.cat (P.P.-O.); jcarratala@bellvitgehospital.cat (J.C.); cgudiol@bellvitgehospital.cat (C.G.); 2Department of Pathophysiology and Transplantation, University of Milan, Via Francesco Sforza, 35, 20122 Milan, Italy; 3Centro de Investigación Biomédica en Red de Enfermedades Infecciosas (CIBERINFEC), Instituto de Salud Carlos III, 28029 Madrid, Spain; cgarciav@clinic.cat; 4Infectious Diseases Department, Hospital Clinic of Barcelona—IDIBAPS, 08036 Barcelona, Spain; puerta@clinic.cat (P.P.-A.); chumbita@clinic.cat (M.C.); 5Departament de Medicina, Facultat de Medicina i Ciències de la Salut, Campus Clínic, Universitat de Barcelona (UB), 08036 Barcelona, Spain; 6Internal Medicine Department, Hospital Reina Sofía de Tudela, 31500 Tudela, Spain; leirehuete@gmail.com; 7Oncology Department, Institut Català d’Oncologia (ICO), Hospital Duran i Reynals, IDIBELL, L’Hospitalet de Llobregat, 08908 Barcelona, Spain; mmarin@iconcologia.net; 8Hematology Department, Institut Català d’Oncologia (ICO), Hospital Duran i Reynals, IDIBELL, L’Hospitalet de Llobregat, 08908 Barcelona, Spain; edomingo@iconcologia.net; 9Centro de Investigación Biomédica en Red de Epidemiología y Salud Pública (CIBERESP), Instituto de Salud Carlos III, 28029 Madrid, Spain; 10Institut Català d’Oncologia (ICO), Hospital Duran i Reynals, IDIBELL, L’Hospitalet de Llobregat, 08908 Barcelona, Spain

**Keywords:** bacteremia, skin and soft-tissue infections, cancer, *Pseudomonas aeruginosa*, antibiotic resistance

## Abstract

Objectives: to assess the current epidemiology, antibiotic therapy and outcomes of onco- hematological patients with bacteremic skin and soft-tissue infections (SSTIs), and to identify the risk factors for Gram-negative bacilli (GNB) infection and for early and overall mortality. Methods: episodes of bacteremic SSTIs occurring in cancer patients at two hospitals were prospectively recorded and retrospectively analyzed. Results: Of 164 episodes of bacteremic SSTIs, 53% occurred in patients with solid tumors and 47% with hematological malignancies. GNB represented 45.5% of all episodes, led by *Pseudomonas aeruginosa* (37.8%). Multidrug resistance rate was 16%. Inadequate empirical antibiotic therapy (IEAT) occurred in 17.7% of episodes, rising to 34.6% in those due to resistant bacteria. Independent risk factors for GNB infection were corticosteroid therapy and skin necrosis. Early and overall case-fatality rates were 12% and 21%, respectively. Risk factors for early mortality were older age, septic shock, and IEAT, and for overall mortality were older age, septic shock and resistant bacteria. Conclusions: GNB bacteremic SSTI was common, particularly if corticosteroid therapy or skin necrosis. IEAT was frequent in resistant bacteria infections. Mortality occurred mainly in older patients with septic shock, resistant bacteria and IEAT. These results might guide empirical antibiotic therapy in this high-risk population.

## 1. Introduction

Skin and soft-tissue infections (SSTIs) are an important cause of morbidity and mortality among hospitalized patients and represent a therapeutic challenge for clinicians. Although uncomplicated SSTIs are often managed successfully on an outpatient basis, more severe infections are not uncommon, including a range of clinical presentations such as deep infections requiring surgical intervention, infections with secondary bacteremia and/or sepsis [1], and infections with local complications such as ischemia or tissue necrosis [2]. In addition, some specific host conditions including immunosuppression, recent surgery, irradiation or diabetes mellitus may lead to more severe presentations and worse outcomes [3].

In immunosuppressed cancer patients, bacterial SSTIs are often accompanied by bacteremia with severe systemic signs and symptoms of infection. Furthermore, neutropenia can reduce or even conceal local signs of infection, and thus complicate the initial evaluation of the clinical presentation [4]. Gram-positives are regarded as the leading cause of SSTIs in the general population. Nevertheless, in immunocompromised patients, such as patients with cancer, and particularly those with neutropenia, Gram-negative bacilli (GNB) may play an important role as causative agents [5].

The last update of the Infectious Diseases Society of America (IDSA) guidelines for the diagnosis and management of SSTIs was published in 2014, almost ten years ago [6]. Since then, the global epidemiology of bacterial infections has changed, with an increase in Gram-negatives and a worrisome emergence of antimicrobial resistance [7]. These developments are of paramount importance in cancer patients, who are particularly prone to infections by resistant bacteria due to repeated cycles of antibiotics, prolonged hospitalizations and immunosuppression [8]. In addition, adequate initial empirical antibiotic therapy is particularly important in this population, because inappropriate empirical antibiotic treatment (IEAT) has been identified as a major risk factor associated with mortality [9,10].

The current IDSA guidelines for the management of patients with febrile neutropenia recommend adding an agent with Gram-positive activity in patients with suspicion of SSTIs [11]. However, at a time when resistant Gram-negatives are emerging as an important cause of bacterial infections in cancer patients, this approach may be insufficient. In addition, the current data on SSTIs in cancer patients are very limited [5,12,13]. Specifically, little is known about potentially severe SSTIs, such as those accompanied by bacteremia and the risk factors for Gram-negative infection. Furthermore, articles including data of a cohort of patients with well-documented microbiological diagnosis of SSTIs, such as those with bacteremia, are missing.

The aim of this study was to assess the currently poorly understood epidemiology, clinical characteristics, empirical antibiotic therapy and outcomes of bacteremic SSTIs in a cohort of immunocompromised cancer patients and hematopoietic stem cell transplant (HSCT) recipients in the current era of antimicrobial resistance, and to identify the risk factors for Gram-negative infection and for mortality.

## 2. Results 

### 2.1. Patient Characteristics and Outcomes

We identified 164 episodes of bacteremic SSTIs in our onco-hematological patients. The epidemiological and clinical characteristics and outcomes of all episodes are summarized in Table 1. The underlying disease was more likely to be the presence of solid tumors than hematological malignancies; breast cancer and lymphoma were the most common conditions in each group of patients. Nineteen patients (12%) were HSCT recipients, in most cases allogeneic HSCT (n = 14). Neutropenia was present in 23% of patients, and 38% received concomitant corticosteroid treatment. The limbs were the most frequent primary location of the infection (59.5%). Thirty-three patients presented skin necrosis (20%), and 57% of the episodes were classified as cSSTI. Appendix A shows the clinical features, therapeutic management and outcomes of patients with necrotizing fasciitis. Septic shock at BSI onset was present in 13% of the patients; 7% required intensive care unit (ICU) admission, and early and overall case-fatality rates were 12% and 21%, respectively.

### 2.2. Etiology and Antibiotic Resistance

The causative organisms of the 164 episodes of bacteremic SSTI are shown in Table 2. Almost half of the isolated pathogens were Gram-positive bacteria, with *S. aureus* being the most frequent (45%), followed by *Streptococcus agalactiae* (13%) and *S. pyogenes* (11%). Among Gram-negative bacteria (46%), *P. aeruginosa* was the most frequently identified (38%), followed by *Escherichia coli* (35%). Twelve episodes were polymicrobial. Overall, 14% of the isolates were MDR. Among GNB, the rate of multidrug resistance reached 16%, with extended-spectrum beta-lactamase production being the most frequent resistance mechanism (n = 8, 61.5%). No carbapenemase production was identified during the study.

In the 88 episodes considered to be cSSTI, the leading cause was *P. aeruginosa* in 20 patients (23%), followed by *E. coli* (18%) and *S. aureus* (17%). 

### 2.3. Risk Factors for Gram-Negative Infection

Table 3 displays the risk factors for Gram-negative infection by univariate and multivariate analysis. Polymicrobial episodes have been excluded to allow the analysis of risk factors for GNB infection to be more homogeneous. The presence of necrosis and the use of concomitant corticosteroids were significantly associated with Gram-negative infection in the univariate and multivariate analyses (OR 2.4, 95% CI 1.03–6, and OR 2.4, 95% CI 1.2–5.1, respectively).

### 2.4. Antibiotic Therapy

The most frequent initial empirical antibiotic therapy was combination therapy (59%), mainly with a cephalosporin plus an aminoglycoside (10%), followed by carbapenem plus glycopeptide (7%). The most frequent monotherapy used was β-lactam + β-lactam inhibitors (50%), followed by cephalosporins (32%) and carbapenems (17%).

The overall rate of inadequate empirical antibiotic therapy (IEAT) was 18%, rising to 35% in patients with infection due to resistant bacteria. IEAT was reported in 29 episodes, including nine cases in whom no empirical antibiotic was started before blood cultures became positive [*S. aureus* (n = 4), MRSA (n = 2) with *Corynebacterium* spp. coinfection in one patient, *S. oralis* (n = 1), *K. pneumoniae* (n = 1) and *Aeromonas hydrophila* (n = 1)]. Therapy was inadequate in the remaining 20 episodes: (i) two patients with candidemia who did not receive antifungal therapy empirically; (ii) two episodes caused by anaerobes (*Clostridium indolis* and *Bacteroides fragilis*); (iii) eight episodes caused by β-lactam-resistant Gram-positive bacteria [MRSA (n = 3), coagulase-negative staphylococci (n = 2), *Corynebacterium* spp. (n = 2) and *E. faecium* (n = 1)]; (iv) five due to MDR-GNB [ESBL-producing enterobacterales (n = 3), Amp-C-producing *K. pneumoniae* (n = 1) and *S. maltophilia* (n = 1); and (v) five episodes which did not receive active agents against the infecting organism [*P. aeruginosa* and piperacillin-tazobactam (n = 3), *E. coli* and amoxicillin/clavulanate (n = 1) and *S. aureus* and ciprofloxacin (n = 1)]. Two of these twenty episodes were polymicrobial.

### 2.5. Risk Factors for Mortality

Table 4 and Table 5 summarize the risk factors associated with early and overall case-fatality rates by univariate and multivariate analyses. Multivariate analysis using a logistic regression model showed that shock at presentation (OR 14.8; 95%CI 3.77–58.5), older age (OR 1.07; 1.02–1.12) and IEAT (OR 5.01; 1.30–19.2) were independently associated with early case-fatality rate. The risk factors for overall case-fatality rate in the multivariate analysis were older age (OR 1.03, 95%CI 1.00–1.06), septic shock at presentation (OR 5.86, 95%CI 1.97–17.4) and infection due to resistant bacteria (OR 2.78, 95%CI 1.01–7.64). IEAT presented a trend toward statistical significance (OR 2.76, 95%CI 0.99–7.69).

## 3. Materials and Methods

### 3.1. Setting, Patients and Study Design

This study was carried out at two university referral hospitals in Barcelona, Spain: the Institut Català d’Oncologia, a 100-bed center for adult cancer patients, and the Hospital Clínic i Provincial, a 1000-bed general hospital. At each institution, all patients with bacteremia are identified daily by the Microbiology laboratory, and visited afterwards by an infectious disease physician who provides medical advice when necessary. Thus, data are prospectively collected in a specific database as part of standard infectious disease management.

We included in the study those adult patients (≥18 years old) with solid tumors, hematological malignancies or HSCT recipients presenting one episode of bacteremic SSTIs from January 2010 to June 2021. The objectives of the study were to assess the etiology, clinical characteristics, empirical antibiotic therapy, and outcomes of onco-hematological patients with bacteremic SSTIs, and to identify the risk factors for Gram-negative infection and for early (7-day) and overall 30-day case fatality rates. For this purpose, we compared the episodes caused by Gram-negatives with those caused by Gram-positives, and we also compared patients who died with those who survived.

The Clinical Research Ethics Committees at both hospitals approved the study. To protect privacy, personal information was encrypted in the electronic database. The need for informed consent was waived by the ethics committees due to the retrospective nature of the data analysis, and because no intervention was carried out.

### 3.2. Definitions

SSTIs were defined as complicated (cSSTIs) if deep subcutaneous tissues were involved and/or surgery was necessary in addition to antimicrobial therapy [14].Necrotizing fasciitis was defined when the infection involved the superficial fascia, the subcutaneous tissue planes and underlying muscles [2].Co-morbidities were defined as the presence of one or more of the following conditions: chronic obstructive pulmonary disease, diabetes mellitus, chronic heart disease, chronic kidney disease (eGFR < 60 mL/min), chronic liver disease and cerebrovascular disease.Neutropenia was defined as an absolute neutrophil count ≤500 cells/μL.Bloodstream infection (BSI) was classified as nosocomial-acquired, healthcare-related, or community-acquired in accordance with previously reported criteria [15].Previous corticosteroid treatment was defined as the administration of 20 mg of prednisone, or equivalent dosing, for at least four weeks within the last 30 days from BSI onset.Septic shock was defined as a systolic blood pressure <90 mmHg that was unresponsive to fluid treatment and that required vasoactive drug therapy.GNB were considered multidrug-resistant (MDR) if any of the following were present: (1) extended-spectrum β-lactamase (ESBL)-producing Enterobacterales, (2) AmpC cephalosporinase hyperproducing Enterobacterales, (3) carbapenem-resistant Enterobacterales, (4) MDR strains of *Pseudomonas aeruginosa* and *Acinetobacter baumannii*, and (5) microorganisms with intrinsic resistance mechanisms (e.g., *Stenotrophomonas maltophilia*).MDR strains were defined based on previously described criteria [16].MDR Gram-positive organisms included methicillin-resistant *Staphylococcus aureus* (MRSA) and ampicillin-resistant *Enterococcus faecium*.Initial empirical antibiotic therapy was considered inadequate if the treatment regimen did not include at least one antibiotic active in vitro against the infecting microorganism.The early and overall case-fatality rates were defined as death from any cause within seven and 30 days of bacteremia onset, respectively.

### 3.3. Microbiology

#### 3.3.1. Blood Sample Collection and Laboratory Equipment

Two sets of two 8–10 mL blood samples (Bactec PlusAerobic and Anaerobic, BD) were taken at 30 min apart from all patients who presented with fever ≥38 °C or when BSI was suspected due to clinical signs or symptoms. Either a BACTEC 9240 apparatus (from 2006 to May 2010) or a BACTEC-FX apparatus (from May 2010, BD Microbiology Systems) were used to process blood samples with an incubation period of five days. Positive blood samples were subcultured onto chocolate agar.

#### 3.3.2. Identification Methods

Identification and antibiotic susceptibility of GNB, *Enterococcus* spp. and *S. aureus* were performed using commercially available MicroScan panels from the Walkaway automated system (Beckman-Coulter). Identification of other *Streptococcus* spp. was performed by standard biochemical testing and antibiotic susceptibility with commercially available Sensititre panels (TREK Diagnostic System). Anaerobes were identified by standard biochemical methods, and antibiotic susceptibility by the E-test method (BioMérieux). From November 2012 onward, identification was also performed by Matrix-Assisted Laser Desorption Ionisation (MALDI-TOF; Bruker Daltonics). Finally, susceptibility or resistance to antimicrobial agents were defined according to the European Committee on Antimicrobial Susceptibility Testing (EUCAST) recommendations [17].

### 3.4. Statistical Evaluation

The Mann–Whitney U test or the Student *t*-test were used to compare continuous variables. Odds ratios (ORs) and 95% confidence intervals (CIs) were calculated, as appropriate. Categorical variables were analyzed by the chi-squared test or Fisher’s exact test using two-by-k contingency tables. All *p*-values were two-tailed and statistical significance was considered at *p* < 0.05. Multivariate conditional logistic regression analysis of factors potentially associated with Gram-negative infection and with mortality included all statistically significant variables in the univariate analysis, together with sex, age, and all clinically relevant variables regardless of whether they were statistically significant.

Analyses were performed by stepwise logistic regression modelling in the SPSS statistics Version 27.0 (SPSS institute Inc., Chicago, IL, USA) [18].

## 4. Discussion

In our large cohort of bacteremic SSTIs in onco-hematological patients, we found that the patients affected were mostly non-neutropenic patients with breast cancer and lymphoma. Gram-negatives remained an important cause of infection, particularly in patients presenting with necrosis and concomitant use of steroids. Mortality occurred mainly in older patients with septic shock at presentation who received IEAT, often due to infection by resistant bacteria.

In our study, the rates of bacteremic SSTI were similar in patients with solid tumors and with hematological malignancies, and only a quarter presented neutropenia. This finding highlights the importance of local factors such as the primary site of the neoplasm with prior surgery and additional local lymphadenectomy and/or radiotherapy in patients with solid tumors, and, to a lesser extent, in patients with lymphoma, as notable risk factors for infection [5]. Other risk factors that predispose to SSTI are the presence of mucositis, epithelial damage and myelosuppression, as well as prolonged use of central venous catheters, skin microbiome dysbiosis, poor wound healing, comorbidities, malnutrition and the use of other immunosuppressants such as corticosteroids [4].

In line with previous studies involving both the general non-immunocompromised population and patients with cancer [4,14,19,20], we found that *S. aureus* was the leading cause of SSTI. However, the rate of Gram-negative infection was particularly high, accounting for almost half of the episodes, rising slightly to 56% in patients with complicated SSTI. Bearing in mind that a large number of patients had breast cancer and that SSTIs in these patients are usually caused by Gram-positives (mainly *S. pyogenes*), we performed a subanalysis of the cohort excluding these patients, and found that the rate of GNB infection remained high (44%).

In addition, more than 15% of the Gram-negative isolates were MDR. These results are in concordance with the increase in Gram-negatives as a leading cause of bacterial infection in cancer patients in recent decades, and with the emergence of antibacterial resistance among them [8,10,21,22]. This is a matter of special concern because the currently available guidelines for the management of SSTI and febrile neutropenia do not include specific recommendations for immunosuppressed cancer patients at risk of MDR-GNB SSTIs. In fact, the IDSA guidelines recommend adding an agent with Gram-positive activity in neutropenic patients with SSTI, but offer no specific guidance for MDR-GNB [11]. In our study, the presence of necrosis and the use of concomitant corticosteroids were significantly associated with an increased risk of GNB infection. To our knowledge, this is the first study evaluating the risk factors for GNB SSTIs in cancer patients.

More than half of the patients presented complicated bacteremic SSTI, including the 30 cases that required drainage or surgery. This percentage is higher than those reported previously in cancer patients and in the general non-immunocompromised population [2,5,6]. In addition, a high proportion of patients debuted with septic shock and presented mortality rates that exceeded the expected figures for this type of infection. This could be explained by the fact that all patients included in our study had concomitant bacteremia, a finding not frequently reported in other studies [5,13].

Five of the seven episodes of necrotizing fasciitis were caused by GNB (two were polymicrobial). Although empirical antibiotic therapy was adequate in almost all patients (86%), the overall case fatality rate reached 43%. Of note, three patients were admitted to the ICU and only three underwent surgery. Necrotizing fasciitis is a life-threatening disease in which surgical treatment is essential to achieving a better prognosis [23,24]. The low rate of surgical debridement in our study is consistent with other reports of lower percentages of surgical treatment among immunocompromised patients compared to their non-immunocompromised peers [12,25]. One plausible explanation for the fact that cancer patients are less likely to undergo surgery could be the fear of the high intraoperative mortality in frail patients with an underlying malignancy land who often present cytopenias.

IEAT was mainly associated with infection due to MDR bacteria, and was associated with an increased risk of death. Since older age and septic shock are not variables in which physicians can intervene, empirical antibiotic therapy remains the only modifiable variable to improve outcomes. In the current era of widespread antimicrobial resistance, updated recommendations for guiding empirical antibiotic therapy in immunocompromised patients with cancer are urgently needed.

This study has some limitations that should be acknowledged. First, some information may have been lost due to its retrospective design, and we may not have adequately controlled for certain confounders. Second, all the patients included in the study had concomitant bacteremia, and so patients with SSTI and no microbiological documentation and those without bacteremia were excluded. Nevertheless, this design allows the identification of patients at higher risk of severe sepsis and worse outcomes, in whom prompt adequate empirical antibiotic treatment and aggressive management are mandatory. Third, the study population may be heterogeneous due to inherent differences in the underlying disease. On the other hand, the main strength of this study is that it describes the largest cohort of cancer patients with bacteremic SSTI, using a multicenter design, even though the participating centers are located in the same geographical area.

In conclusion, we found that bacteremic SSTI due to GNB was common, particularly in patients with concurrent corticosteroid therapy and presenting skin necrosis. *P. aeruginosa* was the leading cause. Mortality occurred mainly in older patients with septic shock at presentation who received inadequate empirical antibiotic therapy, often due to infection by resistant bacteria. In the current era of widespread antimicrobial resistance, updated recommendations for guiding empirical antibiotic therapy in cancer patients with SSTIs are warranted.

## Figures and Tables

**Table 1 antibiotics-12-01722-t001:** Epidemiological and clinical characteristics and outcomes of 164 episodes of bacteremic skin and soft tissue infections.

Characteristic	No.164	Percentage (%)
Age (years, median, range)	61 (19–91)	
Female sex	75	46
Solid tumor	87	53
Breast	16	18
Gynecological	13	15
Colon	10	11.5
Head and neck	9	10
Prostate	7	8
Urothelial	7	8
Other	25	29
Hematological malignancy	77	47
Lymphoma	28	36
Leukemia	25	32
Other	24	31
Hematopoietic stem cell transplant	19	12
Comorbidities	57	35
Diabetes mellitus	26	16
Chronic heart disease	11	7
Chronic obstructive pulmonary disease	9	5.5
Chronic renal disease	7	4
Chronic liver disease	6	4
Primary location SSTI ^a^ (n = 153)		
Limbs	91	59.5
Trunk	33	21.5
Perianal-genital	15	10
Head and neck	14	9
Local infection characteristics		
Erythema	130	79
Pain	115	70
Hot	104	63
Swelling	95	58
Suppuration	66	40
Abscess	27	16.5
Necrosis	33	20
Necrotizing fasciitis	7	4
Complicated SSTI ^a^	88	57
Imaging of affected area (n = 161)	55	
Computed tomography scan	36	22
Ultrasound	13	8
Magnetic resonance	6	4
Microbiological studies		
Culture	57	36
Gram stain	52	33
Site of acquisition		
Health-care	82	50
Nosocomial	44	27
Community	38	23
Previous antibiotic therapy (1 month)	65	40
Neutropenia (<500 neutrophil/µL)	38	23
<100 neutrophil/µL	16	10
Vascular catheter	69	42
Corticosteroid therapy (1 month)	63	38
Septic shock at onset	21	13
Empirical antibiotic therapy (n = 155)		
β-lactam + β-lactam inhibitors	77	50
Cephalosporins	50	32
Carbapenems	26	17
Aminoglycosides	30	19
Glycopeptides	29	19
Quinolones	14	9
Aztreonam	2	1
Daptomycin	20	13
Combination therapy	91	59
Need for drainage/surgery	30	18
Intensive care unit admission	12	7
48 h case-fatality rate	10	6
7-day case-fatality rate	20	12
30-day case-fatality rate	35	21

^a^ SSTI: skin and soft-tissue infection.

**Table 2 antibiotics-12-01722-t002:** Causative organisms of 164 episodes of bacteremic SSTIs in cancer patients and hematopoietic stem cell transplant recipients.

Causative Organisms	No.180 ^a^	Percentage (%)
Gram-positive bacteria	89	49
* Staphylococcus aureus*	40	45
Methicillin-resistant *S. aureus*	8	9
Coagulase-negative staphylococci	5	6
* Streptococcus* spp.	33	37
* S. agalactiae*	12	13
* S. pyogenes*	10	11
* S. pneumoniae*	2	2
* S. equisimilis*	2	2
* S. constellatus*	2	2
* S. anginosus*	2	2
Other *streptococcus* ^b^	3	3
*Enterococcus* spp.	7	8
*Enterococcus faecalis*	4	4.5
*Enterococcus faecium*	3	3
*Corynebacterium* spp.	3	3
*Propionibacterium acnes*	1	1
Gram-negative bacteria	82	46
*Pseudomonas aeruginosa*	31	38
*Escherichia coli*	29	35
*Klebsiella pneumoniae*	11	13
*Enterobacter* spp.	3	4
*Haemophilus influenzae*	1	1
*Morganella morganii*	1	1
*Serratia marcescens*	1	1
*Stenotrophomonas maltophilia*	1	1
*Achromobacter xylosoxidans*	1	1
*Citrobacter* spp.	1	1
*Aeromonas* spp.	1	1
*Moraxella caprae*	1	1
Anaerobic bacteria	7	4
*Bacteroides* spp.	4	
*Clostridium* spp.	2	
*Actinomyces odontolyticus*	1	
Fungi	2	1
*Candida parapsilosis*	1	
*Candida tropicalis*	1	
Polymicrobial infection	12	7
Multidrug-resistant organisms	26	14
Multidrug-resistant GNB ^c^	13	16

^a^ The sum is >164 because 12 episodes were polymicrobial. ^b^
*S. intermedius* (1), *S. oralis* (1) and *S. mitis* (1). ^c^ Multidrug-resistant Gram-negative bacilli: Extended-spectrum beta-lactamase producing- enterobacterales (8), MDR-*P. aeruginosa* (3), MDR-*Achromobacter xyloxosidans* (1), Amp-C producing-*K. pneumoniae* (1).

**Table 3 antibiotics-12-01722-t003:** Risk factors for Gram-negative infection in the univariate and multivariate analyses.

Characteristic	Gram-Positive (N = 77) No. (%)	Gram-Negative (N = 71) No. (%)	Univariate *p*-Value	Adjusted OR (95%CI)	Multivariate *p*-Value
Age, yr; median (range)	64 (19–90)	65 (25–91)	0.875		
Female sex	34 (49)	35 (51)	0.53		
Solid tumor	42 (54)	35 (45)	0.52		
Hematological malignancy	35 (49)	36 (51)	0.52		
Hematopoietic stem cell transplant	8 (10)	10 (14)	0.49		
Co-morbidities	26 (34)	23 (32)	0.86		
COPD ^a^	6(8)	2(3)	0.23		
Diabetes mellitus	12 (16)	11 (15.5)	0.98		
Chronic kidney disease	3 (4)	4 (6)	0.71		
Chronic liver disease	2 (3)	4 (6)	0.43		
Chronic heart disease	2 (3)	7 (10)	0.09		
Previous antibiotic therapy	25 (32.5)	33 (47)	0.07		
Neutropenia (<500 neutrophil/µL)	13 (17)	18 (25)	0.21		
Vascular catheter	27 (35)	34 (48)	0.11		
Corticosteroid therapy (1 month)	24 (31)	33 (47)	0.06	2.4 (1.2–5.1)	0.015
Community-acquired infection	15 (19.5)	19 (27)	0.29		
Shock at presentation	8 (10)	11 (15.5)	0.35		
IEAT ^b^	15 (19.5)	9 (13)	0.26		
Persistent bacteremia	4 (5)	2 (3)	0.68		
Characteristics of the lesion					
Presence of skin necrosis	10 (13)	19 (27)	0.03	2.4 (1.03–6)	0.04
Purulent lesion	28 (36)	35 (49)	0.11		
Necrotizing fasciitis	2 (3)	3 (4)	0.67		
Site of infection					
Trunk	17 (23)	13 (19)	0.54		
Limbs	45 (62)	40 (57)	0.58		
Head and neck	7 (10)	7 (10)	0.93		
Perianal-genital	6 (8)	9 (13)	0.36		
Outcome					
Overall case-fatality rate (30 d)	15 (19.5)	18 (25)	0.39		
Early case-fatality rate (7 d)	9 (12)	10 (14)	0.66		
Very early case-fatality rate (48 h)	4 (5)	6 (8.5)	0.43		

^a^ COPD: chronic obstructive pulmonary disease. ^b^ IEAT: inadequate empirical antibiotic therapy.

**Table 4 antibiotics-12-01722-t004:** Risk factors for 7-day mortality in the univariate and multivariate analyses.

Characteristic	Dead (N = 20)	Alive (N = 144)	Univariate *p*-Value	OR (95%CI)	Multivariate *p*-Value
Age, yr; median (range)	69.5 (56–90)	63 (19–91)	**0.04**	**1.07 (1.02–1.12)**	**0.005**
Female sex	10 (50)	65 (45)	0.68	1.08 (0.32–3.63)	0.89
Solid tumor	13 (65)	75 (52)	0.28		
Hematological malignancy	7 (35)	69 (48)	0.28		
Co-morbidities	7 (35)	50 (35)	1.00		
COPD ^a^	2 (10)	7 (5)	0.30		
Diabetes mellitus	1 (5)	25 (17)	0.20		
Chronic kidney disease	2 (10)	5 (3.5)	0.20		
Chronic liver disease	2 (10)	5 (3.5)	0.20		
Chronic heart disease	1 (5)	10 (7)	1.00		
Gram-negative infection	10 (50)	61 (42)	0.52		
Infection due to MDR ^b^ organisms	7 (35)	19 (13)	0.02	2.4 (0.66–8.84)	0.18
Infection due to MDR-GNB ^c^	3 (15)	10 (7)	0.41		
Neutropenia (<500 neutrophil/µL)	2 (10)	36 (25)	0.17		
Corticosteroid therapy (1 month)	9 (45)	54 (37.5)	0.52		
Shock at presentation	9 (45)	12 (8)	0.00	14.8 (3.77–58.5)	0.000
IEAT ^d^	7 (35)	22 (15)	0.05	5.01 (1.30–19.2)	0.019
Presence of skin necrosis	9 (45)	26 (18)	0.014	2.4 (0.71–8.3)	0.15

^a^ COPD: chronic obstructive pulmonary disease. ^b^ MDR: multidrug resistant. ^c^ GNB: Gram-negative bacilliC. ^d^ IEAT: inadequate empirical antibiotic therapy.

**Table 5 antibiotics-12-01722-t005:** Risk factors for 30-day mortality in the univariate and multivariate analyses.

Characteristic	Dead (N = 35)	Alive (N = 129)	Univariate *p*-Value	OR (95%CI)	Multivariate *p*-Value
Age, yr; median (range)	66 (48–90)	63 (19–61)	0.014	1.03 (1.00–1.06)	0.029
Female sex	18 (51)	57 (44)	0.45	1.33 (0.56–3.18)	0.51
Solid tumor	22 (63)	66 (51)	0.19		
Hematological malignancy	13 (37)	63 (49)	0.19		
Hematopoietic stem cell transplant	2 (6)	17 (13)	0.37		
Co-morbidities	13 (37)	44 (34)	0.74		
COPD ^a^	2 (6)	7 (5)	1.00		
Diabetes mellitus	3 (9)	23 (18)	0.18		
Chronic kidney disease	2 (6)	5 (4)	0.64		
Chronic liver disease	4 (11)	3 (2)	0.059		
Chronic heart disease	2 (6)	9 (7)	1.00		
Community-acquired	8 (23)	29 (22.5)	1.00		
Etiology					
Gram-positive	15 (43)	62 (48)	0.58		
Gram-negative	18 (51)	53 (41)	0.27		
Polymicrobial bacteremia	3 (9)	11 (8.5)	1.00		
Infection due to MDR ^b^ organisms	11 (31)	15 (12)	0.004	2.78 (1.01–7.64)	0.047
Infection due to MDR-GNB ^c^	5 (14)	8 (6)	0.15		
Neutropenia (<500 neutrophil/µL)	7 (20)	31 (24)	0.61		
Corticosteroid therapy (1 month)	19 (54)	44 (34)	0.03		
Surgery/drainage	5 (14)	25 (19)	0.62		
Septic shock at presentation	11 (31)	10 (8)	0.001	5.86 (1.97–17.4)	0.001
IEAT ^d^	10 (29)	19 (15)	0.05	2.76 (0.99–7.69)	0.051
Persistent bacteremia	1 (3)	6 (5)	1.00		
Intensive care unit admission	5 (14)	7 (6)	0.14		
Characteristic of the lesion					
Presence of skin necrosis	13 (37)	22 (17)	0.01	1.89 (0.71–5.04)	0.199
Purulent lesion	16 (46)	55 (43)	0.74		
Necrotizing fasciitis	3 (9)	4 (3)	0.168		

^a^ COPD: chronic obstructive pulmonary disease. ^b^ MDR: multidrug resistant. ^c^ GNB: Gram-negative bacilliC. ^d^ IEAT: inadequate empirical antibiotic therapy.

## Data Availability

The datasets generated and analyzed during the current study are available from the corresponding author upon reasonable request.

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
