# Peer review of "The Etiology, Antibiotic Therapy and Outcomes of Bacteremic Skin and Soft-Tissue Infections in Onco-Hematological Patients"

_antibiotics, 2023, doi:10.3390/antibiotics12121722_

Round 1

Reviewer 1 Report

Comments and Suggestions for Authors

Right from the beginning, I want to congratulate the authors for the study carried out.

Sepsis in patients with neoplasms is a management challenge

Immunosuppression is the main cause of the increased incidence of sepsis in cancer patients

Nevertheless, antibiotic therapy is initiated empirically in these patients as well.

Multidrug resistance in this stage is one of the main factors of unfavorable prognosis. Biomarker-guided Individualization of antibiotic therapy is a method that would contribute to improving the therapeutic strategy in this phase of empirical treatment, before of the culture results . I would like it to be specified  if the only  criteria of diagnosis for IEAT  is if the treatment regimen did not include at least one antibiotic active in vitro against the infecting  microorganism. 

Neutropenia is a negative prognostic marker. I would like to know if the administration of granulocyte transfusions has been studied in patients from the studied groups

Author Response

RESPONSE TO REVIEWERS

Manuscript: antibiotics-2724175

Title: Etiology, antibiotic therapy and outcomes of bacteremic skin and soft-tissue infections in onco-hematological patients

We thank the editor and the reviewers for the constructive remarks, which have helped us to improve the manuscript. All the suggestions have been thoroughly addressed (kindly see the point by point responses below). We highlighted these modifications in an additional copy of the manuscript with tracked changes. Please find below our responses to the reviewers' comments.

REVIEWERS’ COMMENTS TO AUTHORS:

Reviewer#1

Right from the beginning, I want to congratulate the authors for the study carried out.

Sepsis in patients with neoplasms is a management challenge

Immunosuppression is the main cause of the increased incidence of sepsis in cancer patients

Nevertheless, antibiotic therapy is initiated empirically in these patients as well.

Multidrug resistance in this stage is one of the main factors of unfavorable prognosis. Biomarker-guided Individualization of antibiotic therapy is a method that would contribute to improving the therapeutic strategy in this phase of empirical treatment, before of the culture results.

Question #1

I would like it to be specified if the only criteria of diagnosis for IEAT is if the treatment regimen did not include at least one antibiotic active in vitro against the infecting microorganism.

Authors: Thanks for the reviewer’s comments. Yes, in this study we have not considered any other alternative criteria for labeling an empiric antibiotic treatment as inadequate, as mentioned in the Definitions section (lines 156-158).

Question #2

Neutropenia is a negative prognostic marker. I would like to know if the administration of granulocyte transfusions has been studied in patients from the studied groups.

Authors: Despite being considered as a supportive measure in patients with severe bacterial or fungal infection, more or less controversial, it is not a technique that we routinely use in our center and it has not been evaluated in the present study.

Reviewer 2 Report

Comments and Suggestions for Authors

The study investigates the epidemiology, antibiotic therapy, and outcomes of bacteremic skin and soft tissue infections (SSTIs) in onco-hematological patients over an 11-year period. The methodology is well-described, with clear details on patient selection, data collection, and statistical analyses. The identification of risk factors for Gram-negative infection is a valuable contribution, especially the finding of association of necrosis and concomitant corticosteroid use with Gram-negative infection. The study is important for finding risk factors for mortality, mainly the connection between shock at presentation, older age, and inadequate empirical antibiotic therapy and early case-fatality rates, as well as older age, septic shock, and infection due to infection. Overall, the manuscript is well-structured and provides valuable insights. 

Author Response

RESPONSE TO REVIEWERS

Manuscript: antibiotics-2724175

Title: Etiology, antibiotic therapy and outcomes of bacteremic skin and soft-tissue infections in onco-hematological patients

We thank the editor and the reviewers for the constructive remarks, which have helped us to improve the manuscript. All the suggestions have been thoroughly addressed (kindly see the point by point responses below). We highlighted these modifications in an additional copy of the manuscript with tracked changes. Please find below our responses to the reviewers' comments.

REVIEWERS’ COMMENTS TO AUTHORS:

Reviewer#2

The study investigates the epidemiology, antibiotic therapy, and outcomes of bacteremic skin and soft tissue infections (SSTIs) in onco-hematological patients over an 11-year period. The methodology is well-described, with clear details on patient selection, data collection, and statistical analyses. The identification of risk factors for Gram-negative infection is a valuable contribution, especially the finding of association of necrosis and concomitant corticosteroid use with Gram-negative infection. The study is important for finding risk factors for mortality, mainly the connection between shock at presentation, older age, and inadequate empirical antibiotic therapy and early case-fatality rates, as well as older age, septic shock, and infection due to infection. Overall, the manuscript is well-structured and provides valuable insights. 

Authors: We deeply appreciate the reviewer's comment.  

Reviewer 3 Report

Comments and Suggestions for Authors

Etiology, Antibiotic Therapy, And Outcomes Of Bacteremic Skin And Soft-Tissue Infections In Oncohematological Patients

Overall, this is a valuable contribution to the literature on SSTIs in oncohematological patients. The study provides important insights into the epidemiology, microbiology, and risk factors for these infections in this high-risk patient group. The study recruited a convenient cohort size representing BSI/SSTIs in oncohematological patients.

General Comments:

§  Ensure that the language is consistent and grammatically correct throughout the article.

§  Some sentences are overly wordy and could benefit from simplification and brevity.

§  Consider breaking long sentences into shorter ones to improve clarity & readability.

Abstract:

§  Elaborate further on the methodology, incorporating additional information regarding patient selection, data collection, and statistical analyses. Incorporate the study's duration to offer context for the findings. Also, incorporate details about the participating hospitals pertinent information.

Introduction:

§  Line 50-52: Provide relevant references to strengthen the scientific validity of the statement. [In addition, some specific host conditions including  immunosuppression, recent surgery, irradiation, or diabetes mellitus may lead to more  severe presentations and worse outcomes.]

§  Line 56-59: Provide relevant references to strengthen the scientific validity of the statement. [Gram-positives are regarded as the leading cause of SSTIs in the general population. Nevertheless, in immunocompromised patients such as patients with cancer, and particularly those with neutropenia, Gram-negative bacilli (GNB) may play an important role as causative agents.]

§  Line 70-75: The introduction discusses the limitations of the current data on SSTIs in cancer patients but could be more explicit in stating the research gap that this study aims to address.

§  Line 76-80: You might consider rephrasing the objectives of the study section for a more precise and impactful statement of purpose, and emphasize the novelty of the study.

Materials and methods

§  Line 87-91: This statement may give the impression that your study focused on bacteremia (Blood-stream infections), rather bacteremia on top of SSTIs. To clarify your research focus, consider rephrasing it to emphasize that your study investigated bacteremia secondary to complicated SSTIs. [All patients with bacteremia were identified daily by the Microbiology laboratory, and visited afterward by an infectious disease physician who provided medical advice when necessary. All consecutive episodes of bacteremic SSTIs from January 2010 to June 2021 were analyzed.]

§  Replace the term "bacteremic SSTIs" with "bacterial SSTIs" throughout the manuscript to avoid confusion with pure bloodstream infections. You can utilize an acronym such as "BSI/SSTI" to symbolize the combination of bloodstream infections on top of Skin and Soft Tissue Infections.

§  Include a statement that distinctly outlines the criteria for including and excluding participants in the study cohort. E.g.: age category, confirmed diagnosis of hematological malignancies, bacterial complicated skin and soft-tissue infections, and patients with a recorded absolute neutrophil count at the onset of bacteremia.

§  Definitions are provided for various terms, but consider using bullet points or a table to make them more visually accessible.

§  Microbiology Procedures: consider breaking this section into subsections for better organization and clarity. For example, you could have subsections for blood sample collection, laboratory equipment, and identification methods.

§  Line 106-111: Define co-morbidities like chronic kidney disease, and chronic liver disease in terms of laboratory findings (creatinine clearance, AST, and ALT levels. Explicitly define nosocomial-acquired, healthcare-related, or community-acquired.

§  At the first instance of use, spell out the abbreviation "BSI". There seems to be occasional interchangeability with "SSTIs" in the manuscript, please clarify!!!!

§  Please provide more detail on what variables were considered in the statistical analysis and how they were chosen to be included in both univariate and multivariate analyses.

Results:

§  Table 1 & 2: According to the result section, 164 episodes of BSI/SSTIs were identified. Table 1 presents a summary of the study cohort's demographics, while Table 2 provides details of 180 episodes (Actual numbers indicated in the table: Gram-positive 89, Gram-negative 92, Anaerobes 7, and Fungi 1 = Total 179). The authors should investigate the source of the discrepancies in numbers all over the manuscript.

§  Table 3: again different number (compared to Table 2) in each bacterial category was identified, Gram-positive (N=77) Gram-negative (N=71). The authors should investigate the source of the discrepancies in numbers all over the manuscript.

§  Instead of providing a selected parameter, I prefer to mention the P values of all investigated parameters mentioned in the tables in all statistical modules.

§  Propensity scores of case severity (mutagenicity & comorbidity) would be an additional value if could be included in this study.

Comments on the Quality of English Language

§  Ensure that the language is consistent and grammatically correct throughout the article.

§  Some sentences are overly wordy and could benefit from simplification and brevity.

§  Consider breaking long sentences into shorter ones to improve clarity & readability.

Author Response

RESPONSE TO REVIEWERS

Manuscript: antibiotics-2724175

Title: Etiology, antibiotic therapy and outcomes of bacteremic skin and soft-tissue infections in onco-hematological patients

We thank the editor and the reviewers for the constructive remarks, which have helped us to improve the manuscript. All the suggestions have been thoroughly addressed (kindly see the point by point responses below). We highlighted these modifications in an additional copy of the manuscript with tracked changes. Please find below our responses to the reviewers' comments.

REVIEWERS’ COMMENTS TO AUTHORS:

Reviewer#3

Overall, this is a valuable contribution to the literature on SSTIs in oncohematological patients. The study provides important insights into the epidemiology, microbiology, and risk factors for these infections in this high-risk patient group. The study recruited a convenient cohort size representing BSI/SSTIs in oncohematological patients.

General Comments and on the Quality of English Language:

  • Ensure that the language is consistent and grammatically correct throughout the article.
  • Some sentences are overly wordy and could benefit from simplification and brevity.
  • Consider breaking long sentences into shorter ones to improve clarity & readability.

Authors: According to the reviewer suggestion, we have reviewed the manuscript and we have identified some typographical errors that have been corrected appropriately. In addition, the manuscript was sent to the University Language Department for English editing before submission.

Abstract

Question #1

Elaborate further on the methodology, incorporating additional information regarding patient selection, data collection, and statistical analyses. Incorporate the study's duration to offer context for the findings. Also, incorporate details about the participating hospitals pertinent information.

Authors: Although we would certainly like to be able to develop the aforementioned sections further, the limitation of 200 words maximum makes it difficult to fit all this information in the abstract. In this case, we have prioritized further developing other sections, but we hope to have been able to provide a more extensive description throughout the manuscript.

Introduction

Question #1

Line 50-52: Provide relevant references to strengthen the scientific validity of the statement. [In addition, some specific host conditions including immunosuppression, recent surgery, irradiation, or diabetes mellitus may lead to more severe presentations and worse outcomes.]

Authors: In accordance with the reviewer’s request, we have added a suitable reference to strengthen the scientific validity of the mentioned statement [3. Ki V, Rotstein C. Bacterial skin and soft tissue infections in adults: A review of their epidemiology, pathogenesis, diagnosis, treatment and site of care. Can J Infect Dis Med Microbiol. 2008 Mar;19(2):173-84. doi: 10.1155/2008/846453].

Question #2

Line 56-59: Provide relevant references to strengthen the scientific validity of the statement. [Gram-positives are regarded as the leading cause of SSTIs in the general population. Nevertheless, in immunocompromised patients such as patients with cancer, and particularly those with neutropenia, Gram-negative bacilli (GNB) may play an important role as causative agents.]

Authors: Since the old reference number 11 explicitly addresses this issue, we have modified its order (now number 5) to correctly link it to the statement.

Question #3

Line 70-75: The introduction discusses the limitations of the current data on SSTIs in cancer patients but could be more explicit in stating the research gap that this study aims to address.

Authors: Following the reviewer’s advice, we have included some sentences in the text that highlight the research gap we aim to address (lines 80-83).

Question #4

Line 76-80: You might consider rephrasing the objectives of the study section for a more precise and impactful statement of purpose, and emphasize the novelty of the study.

Authors: According to the reviewer, we have added some nuances to emphasize the contribution of this study (line 84).

Materials and methods

Question #1

Line 87-91: This statement may give the impression that your study focused on bacteremia (Blood-stream infections), rather bacteremia on top of SSTIs. To clarify your research focus, consider rephrasing it to emphasize that your study investigated bacteremia secondary to complicated SSTIs. [All patients with bacteremia were identified daily by the Microbiology laboratory, and visited afterward by an infectious disease physician who provided medical advice when necessary. All consecutive episodes of bacteremic SSTIs from January 2010 to June 2021 were analyzed.]

Authors: Thanks for the reviewer’s remark. We have rephrased the aforementioned statement in order to clarify that we routinely collected data about all bacteremia episodes and, afterwards, we selected and analyzed those episodes secondarily occurring due to SSTIs for the present study.

Question #2

Replace the term "bacteremic SSTIs" with "bacterial SSTIs" throughout the manuscript to avoid confusion with pure bloodstream infections. You can utilize an acronym such as "BSI/SSTI" to symbolize the combination of bloodstream infections on top of Skin and Soft Tissue Infections.

Authors: We truly believe that the term “bacteremic SSTIs” is quite accurate for the purpose of the study, which includes patients with skin and soft-tissue infections with also secondary bacteremia. The term “bacterial SSTIs” would not be a proper category because it only refers to bacterial infection, regardless of whether patients are bacteremic or not. Likewise, in our opinion, the acronym “BSI/SSTI” might lead to confusion because it is not clear whether both conditions must occur separately. Furthermore, we have used similar terms (bacteremic pneumonia and bacteremic cholangitis) in previous published articles with no inconveniences [Albasanz-Puig A et al., On Behalf Of The Ironic Study Group. Effect of Combination Antibiotic Empirical Therapy on Mortality in Neutropenic Cancer Patients with Pseudomonas aeruginosa Pneumonia. Microorganisms (2022); Royo-Cebrecos C et al., Characteristics, aetiology, antimicrobial resistance and outcomes of bacteraemic cholangitis in patients with solid tumours: A prospective cohort study. J Infect (2017)].

Question #3

Include a statement that distinctly outlines the criteria for including and excluding participants in the study cohort. E.g.: age category, confirmed diagnosis of hematological malignancies, bacterial complicated skin and soft-tissue infections, and patients with a recorded absolute neutrophil count at the onset of bacteremia.

Authors: As the reviewer suggested, we have included a statement that refers to the study inclusion criteria (lines 98-100).

Question #4

Definitions are provided for various terms, but consider using bullet points or a table to make them more visually accessible.

Authors: Bullet points have been used in the Definitions section, according to the reviewer.

Question #5

Microbiology Procedures: consider breaking this section into subsections for better organization and clarity. For example, you could have subsections for blood sample collection, laboratory equipment, and identification methods.

Authors: Microbiology section has been properly broken into two subsections, according to the reviewer’s suggestion.

Question #6

Line 106-111: Define co-morbidities like chronic kidney disease, and chronic liver disease in terms of laboratory findings (creatinine clearance, AST, and ALT levels. Explicitly define nosocomial-acquired, healthcare-related, or community-acquired.

Authors: We have defined chronic kidney disease as eGFR<60ml/min according to the KDIGO CKD classification. However, it is not common to define chronic liver disease in terms of laboratory findings, given that ALT/AST levels do not correlate well. On the other hand, the concepts nosocomial-acquired, healthcare-related, and community-acquired are already defined in the attached reference. In order to avoid unnecessary information in the text, we would rather maintain the respective reference for these well-known definitions.

Question #7

At the first instance of use, spell out the abbreviation "BSI". There seems to be occasional interchangeability with "SSTIs" in the manuscript, please clarify!!!!

Authors: Thanks to allow us to correct this mistake. We have already fixed it accordingly.

Question #8

Please provide more detail on what variables were considered in the statistical analysis and how they were chosen to be included in both univariate and multivariate analyses.

Authors: As mentioned in the Statistical Evaluation section, the considered variables in the statistical analysis were those factors potentially associated with Gram-negative infection and with mortality. The chosen variables for univariate analyses appear on tables 3, 4 and 5. Multivariate analyses included all statistically significant variables in the univariate analysis, together with sex, age, and all clinically relevant variables regardless of whether they were statistically significant (lines 217-221).

Results:

Question #1

Table 1 & 2: According to the result section, 164 episodes of BSI/SSTIs were identified. Table 1 presents a summary of the study cohort's demographics, while Table 2 provides details of 180 episodes (Actual numbers indicated in the table: Gram-positive 89, Gram-negative 92, Anaerobes 7, and Fungi 1 = Total 179). The authors should investigate the source of the discrepancies in numbers all over the manuscript.

Authors: The mentioned numerical discrepancy relies on the 12 polymicrobial episodes (³ 2 isolates), as now clarified in the footer of Table 2. Likewise, there are two episodes of candidemia instead of one.

Question #2

Table 3: again, different number (compared to Table 2) in each bacterial category was identified, Gram-positive (N=77) Gram-negative (N=71). The authors should investigate the source of the discrepancies in numbers all over the manuscript.

Authors: Polymicrobial episodes have been excluded to allow the analysis of risk factors for GNB infection to be more homogeneous.

Question #3

Instead of providing a selected parameter, I prefer to mention the P values of all investigated parameters mentioned in the tables in all statistical modules.

Authors: Tables 1 and 2 do not contain P values because they only provide descriptive analysis. Tables 3, 4 and 5 already provide P values for all the variables of interest.

Question #4

Propensity scores of case severity (mutagenicity & comorbidity) would be an additional value if could be included in this study.

Authors: Thank you for your comment. However, we believe that including a propensity score analysis in this study would not provide any benefit, because we are not comparing two specific cohorts of patients.

Reviewer 4 Report

Comments and Suggestions for Authors

The manuscript “Etiology, antibiotic therapy and outcomes of bacteremic skin and soft-tissue infections in onco-hematological patients” is well written and data clearly presented.

Minor comments:

In table 3,4, and 5 title, indicate the multivariate and univariate p-value instead of just “p” for both.

Author Response

RESPONSE TO REVIEWERS

Manuscript: antibiotics-2724175

Title: Etiology, antibiotic therapy and outcomes of bacteremic skin and soft-tissue infections in onco-hematological patients

We thank the editor and the reviewers for the constructive remarks, which have helped us to improve the manuscript. All the suggestions have been thoroughly addressed (kindly see the point by point responses below). We highlighted these modifications in an additional copy of the manuscript with tracked changes. Please find below our responses to the reviewers' comments.

REVIEWERS’ COMMENTS TO AUTHORS:

Reviewer#4

The manuscript “Etiology, antibiotic therapy and outcomes of bacteremic skin and soft-tissue infections in onco-hematological patients” is well written and data clearly presented.

Minor comments:

In table 3,4, and 5 title, indicate the multivariate and univariate p-value instead of just “p” for both.

Authors: According to the reviewer’s request, we have indicated it accordingly.

Round 2

Reviewer 3 Report

Comments and Suggestions for Authors

I sincerely appreciate the edits provided by the authors who have addressed each comment that aimed to strengthen the article's clarity.

I'm confident that the revisions have enhanced the overall quality of the manuscript, and I find no objections to publishing it in its current form.

Regards